# Ferritin Heavy Chain Binds Peroxiredoxin 6 and Inhibits Cell Proliferation and Migration

**DOI:** 10.3390/ijms232112987

**Published:** 2022-10-26

**Authors:** Maddalena Di Sanzo, Flora Cozzolino, Anna Martina Battaglia, Ilenia Aversa, Vittoria Monaco, Alessandro Sacco, Flavia Biamonte, Camillo Palmieri, Francesca Procopio, Gianluca Santamaria, Francesco Ortuso, Piero Pucci, Maria Monti, Maria Concetta Faniello

**Affiliations:** 1Research Center of Biochemistry and Advanced Molecular Biology, Department of Experimental and Clinical Medicine, “Magna Graecia” University of Catanzaro, 88100 Catanzaro, Italy; 2Department of Chemical Sciences, Università degli Studi di Napoli “Federico II”, Via Cinthia 21, 80126 Napoli, Italy; 3CEINGE Biotecnologie Avanzate, Via G. Salvatore 486, 80145 Napoli, Italy; 4Interdepartmental Centre of Services, “Magna Graecia” University of Catanzaro, 88100 Catanzaro, Italy; 5Department of Health Science, “Magna Graecia” University of Catanzaro, 88100 Catanzaro, Italy

**Keywords:** H Ferritin subunit, PRDX6, protein-protein interaction

## Abstract

The H Ferritin subunit (FTH1), as well as regulating the homeostasis of intracellular iron, is involved in complex pathways that might promote or inhibit carcinogenesis. This function may be mediated by its ability to interact with different molecules. To gain insight into the FTH1 interacting molecules, we analyzed its interactome in HEK293T cells. Fifty-one proteins have been identified, and among them, we focused our attention on a member of the peroxiredoxin family (PRDX6), an antioxidant enzyme that plays an important role in cell proliferation and in malignancy development. The FTH1/PRDX6 interaction was further supported by co-immunoprecipitation, in HEK293T and H460 cell lines and by means of computational methods. Next, we demonstrated that FTH1 could inhibit PRDX6-mediated proliferation and migration. Then, the results so far obtained suggested that the interaction between FTH1/PRDX6 in cancer cells might alter cell proliferation and migration, leading to a less invasive phenotype.

## 1. Introduction

Ferritin, the major intracellular iron-storage protein, binds iron in a soluble, non-toxic form and makes it available for many cellular processes. It is localized in cytoplasm [1], nucleus [2] and mitochondria [3] and is composed of 24 subunits of two different types, a light chain (L; FTL) 19 kDa and a heavy chain (H; FTH1) 21 kDa. The two subunits, which share extensive homology, are functionally distinct: FTH1 is responsible for the ferroxidase activity of the ferritin molecule, while FTL is mainly involved in iron storage [4,5]. FTH1 and FTL are coded by two different genes [6,7] whose activity is regulated at transcriptional [8,9,10] and post-transcriptional levels. The post-transcriptional control of the ferritin genes is iron-dependent [11] and acts mainly on the translational efficiency of the FTH1 and FTL mRNAs [12]. FTH1 controls the cellular pool of chelatable and redox-active iron, named the labile iron pool (LIP) [13]. It has been shown that FTH1 repression evokes an increase in the cellular LIP level and activity, while its overexpression decreases the LIP content [14]. A high level of LIP results in oxidative damage by catalyzing ROS generation in mitochondria [15]. Besides its role in iron metabolism, FTH1 is involved in signaling pathways related to physiologic and pathologic processes by interacting with different protein partners. FTH1 regulates angiogenesis during inflammation and malignancy by antagonizing the cleavage of HKa (high molecular weight kininogen), an endogenous inhibitor of angiogenesis [16], which is also involved in the onset of HIV-mediated neuropathology in patients with a history of drug abuse [16]. FTH1 interacts with the Alacrima-Acalasia-Adrenal Insufficiency Neurological Disorder (ALADIN) protein [17], whose mutations determine the triple A syndrome characterized by impaired FTH1 nuclear uptake [17]. FTH1 may also physically interact with the CXCR4 chemokine receptor, and in response to CXCL12 stimulation, FTH1 is phosphorylated and translocated into nuclei [18]. FTH1/CXCR4 interaction also has functional feedback because CXCR4-mediated ERK1/2 activation is inhibited by FTH1 overexpression [18]. FTH1, under oxidative stress, interacts with p53, thus increasing its transcriptional activity [19]. Moreover, it also binds Nuclear Receptor Coactivator-4 (NCOA4), which targets its degradation to iron recycling [20]. Furthermore, FTH1 may form a complex with death domain-associated nuclear proteins (Daxx), and together they can participate in the apoptosis pathway [21]. FTH1 might function as a tumor suppressor or as an oncogene; indeed, FTH1 knockdown reduces, both in vivo and in vitro, melanoma cell proliferation [22] and may modulate the MHC class I molecules expression leading to NK cells activation [23] while in the erythroleukemia K562 cell line determines an increased expression of a specific set of onco-miRNAs [24], activation of H19/miR-675 axis [25] and severe protein misfolding [26]. In the MCF7 and in NCI-H460 cells induced epithelial-to-mesenchymal transition (EMT) [27], while in ovarian cancer cells, FTH1 is involved in the inhibition of cancer cell proliferation and in cancer stem cell propagation [28]. The evidence that FTH1 may physically interact with molecules expressed in various human malignancies raises the question of the functional role of FTH1 in critical cellular pathways. This function may be mediated by its ability to interact with several signaling molecules. Therefore, studies focused on FTH1 “interactome” analysis are expected to allow an understanding of both physiological and pathological processes involving it. The investigation of new potential FTH1 interacting proteins was performed by using a functional proteomic approach, relying on the immunoprecipitation of a 3xFlag-FTH1 containing complexes from HEK-293T protein extract and their identification by nanoLC-MS/MS methodologies. In this study, we identified Prdx6 as a novel FTH1 interacting protein and that both were involved in the cellular response to oxidative stress. We demonstrated that the FTH1/PRDX6 interaction also occurred in NCI-H460 cells and investigated the functional role of this binding in cell migration and proliferation.

## 2. Results

### 2.1. Identification of FTH1 Interacting Proteins

To unravel novel functional roles of ferritin-heavy chain (FTH1), its interactome has been investigated in order to identify new putative protein partners by immunoprecipitation of a tagged 3xFlag-FTH1 and its interacting proteins from HEK-293T protein extracts. HEK-293T cells, transfected with an empty vector, were used as control. Immunoprecipitated proteins were fractionated by SDS-PAGE, and the 39 slices cut from both sample and control lanes (Appendix A) were in situ digested by trypsin. Peptide mixtures were then analyzed by LC-MS/MS, and proteins were identified by Mascot software.

The final list of FTH1’s putative partners, obtained by removing common proteins between the sample and control, is reported in Appendix A.

Subsequently, a functional analysis of identified FTH1 putative interactors was conducted according to the STRING database, up to a maximum of five co-interactors.

The graphical representation of the inferred network (Figure 1), weighted by protein interactions, provided a detailed view of the putative functional linkage among the 51 proteins used as input and FTH1, facilitating the comprehension of the modularity analysis in biological processes. The analysis generates a putative protein–protein interaction score. We sorted interactions by applying a score threshold of 0.3 and being FTH1 interactors. 

The majority of FTH1 interacting proteins belongs to the cytoskeleton, cell proliferation and traffic, signal transduction, spindle organization and ferroptosis process. Among the proteins already known to interact with FTH1, NCOA4 has been identified, and the interaction has also been confirmed by immunoprecipitation (data not shown). Among the novel putative FTH1 interacting proteins was identified the HSPB1 protein involved in iron metabolism. HSPB1inhibits transferrin receptor 1 (TFR1) expression by modulating intracellular iron accumulation. CDKAL1 belongs to the methyltransferase family, whose function is impaired by cellular iron deficiency. Another interactor is Vimentin, whose function is also modulated by iron metabolism. 

Additionally, a novel partner, Peroxiredoxin 6 (PRDX6), captured our attention, being co-expressed with FTH1 in different tissue (see Appendix A) and involved in several biological processes, including focal adhesion (FDR 1.35 × 10^−5^, cell junction (FDR 0.006), signaling by ROBO receptors (FDR 0.01) and activation of ATR in response to replication stress (FDR 0.02), as predicted by the STRING functional association analysis. (Appendix A). PRDX6 is an enzyme involved in the containment of oxidative stress preventing cell peroxidation causative of membrane lesions [29]. Moreover, it has been demonstrated that its catalytic activity favors cancer cell proliferation, motility promoting invasion and metastasis processes [30,31]. In light of these important roles in promoting pathological states, the interaction between FTH1 and PRDX6 was further investigated.

### 2.2. FTH1/PRDX6 Interaction in HEK-293T and NCI-H460 Cells

To assess PRDX6-FTH1 physical interaction, HEK-293T cells transiently transfected with 3xFlag-FTH1 or 3xFlag control vector were employed for the co-immunoprecipitation experiment. The cell lysates were incubated with the ANTI-FLAG M2 affinity gel, and the immunoprecipitated complexes were separated by SDS-PAGE and analyzed by Western blot with anti-PRDX6 and anti-Flag antibodies, respectively. As shown in Figure 2A, the interaction between the two proteins was confirmed. The same result was obtained by investigating the interaction between the endogenous proteins expressed at basal levels in wild-type HEK-293T cells (Figure 2B). Confocal microscopy analysis demonstrated the cytoplasmatic colocalization of FTH1 and PRDX6 (Figure 2C), as expected by their localization. Furthermore, to verify if the interaction between FTH1 and PRDX6 is independent by cell lines, we replicated the same experiment in non-small cell lung cancer (NSCLC) NCI-H460 cells, transiently transfected with 3xFlag-FTH1 or 3xFlag control vector. In NCI-H460 cells, as well as in HEK-293T, FTH1 interacted with PRDX6 (Figure 3A) and colocalized in the cytoplasm (Figure 3B).

### 2.3. Molecular Recognition in FTH1/PRDX6 Complex

The interaction between FTH1 and PRDX6 was investigated by means of computational methods. Theoretical models of FTH1 and PRDX6, the latter analyzed both as reduced (PRDX6_r) and sulphinic acid (PRDX6_s) states, were built and submitted to docking simulation (as reported in Materials and Methods). Results clearly indicated a productive recognition of FTH1 against both models of PRDX6 states. Globally, 50 and 55 possible configurations of FTH1 were predicted against PRDX6_r and PRDX6_s targets, respectively. The average interaction energies were very similar and were estimated at −15.44 kcal/mol for FTH1•PRDX6_r and at −15.59 kcal/mol for FTH1•PRDX6_s. Even considering the most stable configuration only, the difference was lower than 1 kcal/mol. All docking-generated complexes were graphically inspected, revealing a wide exploration of FTH1 around both targets (Figure 4). 

On the theoretical complexes, the in-house GBPM method was applied to highlight and classify, by quartiles, the most relevant interacting residues (Table 1).

Interestingly, in both PRDX6 models, FTH1 was able to directly recognize several residues close to His39, Cys47 and Arg132, corresponding to the PRDX6 peroxidase site, and to His26, Ser32 and Asp140 that are involved in the PRDX6 phospholipase activity (Table 1). These interactions could rationalize the experimentally observed FTH1 modulation of PRDX6. To deeply investigate the above-reported event, all theoretical complexes were searched to identify, among the highest complexes stabilizing ones, the FTH1 residues recognizing PRDX6 phospholipase or peroxidase sites. Of course, the phospholipase site, with more solvent exposed than the peroxidase one, allowed larger interaction with FTH1 (Figure 5). 

### 2.4. FTH1 Inhibits Proliferation and Migration in PRDX6 Overexpressed NCI-H460 Cells

Accumulated evidence suggests that PRDX6 exerts specific functions in cancer progression, affecting cell growth, survival, migration, differentiation, invasiveness, and metastasis. In NSCLC, PRDX6 over-expression promotes cancer cell proliferation and invasive phenotype [30]. To investigate whether the interaction between FTH1 and PRDX6 might affect cell migration and proliferation, NCI-H460 cells were transiently transfected with 3xFlag-PRDX6, 3xFlag-PRDX6/3xFlag-FTH1, 3xFlag-PRDX6/siFTH1 or 3xFlag control vector. The confluent NCI-H460 cells were scraped to create a wound, and cell migration was assessed 24, 48 and 72 h later. The results of a triplicate set of independent assays (A and B of Figure 6) demonstrate that at 72 h after the scratch, the wound area of 3xFlag-PRDX6 cells was significantly narrower than that of the control. Interestingly, when FTH1 and PRDX6 were contemporaneously overexpressed, the wound area was significantly larger than in the presence of PRDX6 alone. Conversely, silencing of FTH1 induced by siRNA transfection and PRDX6 overexpression significantly increased the migratory ability compared to PRDX6 and FTH1 overexpressed cells. Next, we compared the proliferation ability of NCI-H460 FTH1/PRDX6 vs. NCI-H460 PRDX6 and vs. NCI-H460 siFTH1/PRDX6 cells. The results of the MTT test, reported in Figure 6C, showed that, at 72 h, the simultaneous overexpression of FTH1 and PRDX6 was able to significantly reduce cell proliferation with respect to the PRDX6 overexpression, the FTH1 knockdown and PRDX6 overexpression increased proliferation rate compared to FTH1 and PRDX6 overexpression. Western blot analysis (Figure 6D) demonstrated that FTH1 levels are markedly reduced in 3xFlag-PRDX6 cells compared to both control and 3xFlag-FTH1/3xFlag-PRDX6 overexpression. These data strongly indicate that the presence of FTH1 might counteracts cell proliferation and migration processes induced by PRDX6 overexpression.

## 3. Discussion

In recent years, our work has mainly focused on the analysis of the iron-independent roles of FTH1 in human-transformed cell lines [22,27,28,32]. Taken all together, the data from our and other groups strongly suggest that the FTH1 is involved in the complex pathways that might promote or inhibit carcinogenesis [33]. Although a lot of experimental evidence suggests that altered expression of FTH1 is a common event in cancer, most of the available studies do not clarify the specific role of FTH1 in this process. This is mainly due to incomplete knowledge of the overall scenario in which FTH1 and its partners interact to promote tumor progression. Therefore, the knowledge of molecules interacting with FTH1 can clarify this issue. Identification and characterization of each component of protein networks is a critical step to fully understand these processes at the molecular level. 

The aim of this paper was the identification of FTH1 protein partners to further shed light on its functions in cancer cells. To this purpose, we have pursued a proteomic-based approach relying on the IP-MS (Immunoprecipitation—Mass Spectrometry) of 3xFLAG-FTH1 protein overexpressed in HEK-293T cells. 

In order to further investigate the FTH1 role in tumorigenesis, we focused our attention on the PRDX6, a member of the peroxiredoxin family. PRDX6 is a bifunctional enzyme and possesses both peroxidase and calcium-independent phospholipase A2 (iPLA2) activity, protects against oxidative stress and prevents cell peroxidation causative of membrane lesions [29]. Elevated levels of PRDX6 expression have been shown to be associated with a variety of human cancers, among which lung and breast [30,34]. In lung cancer cells, it has been demonstrated that the upregulation of PRDX6 results in the activation of Akt via phosphoinositide 3-kinase (PI3K) and p38 kinase [35]. It has been demonstrated that PRDX6 promotes cell proliferation and inhibits cancer cell apoptosis; its peroxidase activity promotes the growth of cancer cells, whereas its PLA2 activity promotes the invasion and metastasis of cancer cells [30]. Their physical interaction was also confirmed by co-immunoprecipitation in both FTH1 overexpressing cells and in basal conditions. Confocal images further confirmed the interaction between FTH1 and PRDX6 and showed their cytoplasmic predominantly colocalization. Our data, in agreement with the literature, showed that PRDX6 overexpression induced NCI-H460 cell proliferation and migration; on the contrary, NCI-H460 cells show a strong reduction of their proliferative and migratory capability when simultaneous transfected with FTH1 and PRDX6. These results demonstrate that the interaction between FTH1/PRDX6 alters cell proliferation and migration leading to a less invasive phenotype, suggesting for FTH1 an inhibitory effect on PRDX6 oncogenic role.

It has been demonstrated that PRDX6 is highly expressed in human cancer cells and plays a critical role in cancer development [36,37]. Indeed, high levels of PRDX6 contribute to the proliferation of cancer cells [30]. So, our findings demonstrate the inhibitory effects of FTH1 on PRDX6 and suggest a new potential therapeutic target for the development of novel therapeutic strategies for the treatments of such cancers overexpressing PRDX6.

## 4. Materials and Methods

### 4.1. Cell Cultures

HEK-293T cells (Sigma-Aldrich, St. Louis, MO, USA) human embryonic kidneys were cultured in adherent conditions in a DMEM (Sigma-Aldrich) medium with 10% FBS and 1% Penicillin-Streptomycin (Sigma-Aldrich). NCI-H460 human non-small lung cancer cells (ATCC, Manassas, VA, USA) were cultured in an RPMI 1640 (Sigma-Aldrich) medium supplemented with 10% fetal bovine serum (FBS, Belize City, Belize) and 1% Penicillin-Streptomycin (Sigma-Aldrich). The two cell lines were maintained at 37 °C in a humidified 5% CO_2_ atmosphere.

### 4.2. Transient Transfection and Cell Lysis

HEK-293T cells were transiently transfected with an expression vector containing the full length of human FTH1 cDNA (3xFlag-FTH1) (N-Flag tag-Sino Biological, Beijing, China), using Lipofectamine 3000 transfection reagent (Thermo Fisher Scientific, Waltham, MA, USA) following the manufacturer’s instructions [38]. Cells transfected with empty vector 3xFlag were used as a negative control. Cells were incubated in the medium containing the transfection mixture for up to 72 h. Experiments were performed at least three times. Cell lysis was conducted using a buffer composed of the following components: 50 mM Tris/HCl pH 7.4, 150 mM NaCl, 10% Glycerol, 1 mM Na_3_(VO)_4_, 1 mM NaF, 0, 5 mM PMSF, 1% Triton and 1 mini pill of protease inhibitor (EDTA free ROCHE). Cell pellets were resuspended, left for 10 min in ice, and then put on the tube rotator at 4 °C for 45 min. Finally, the lysates were centrifuged for 30 min at 13,000 rpm and 4 °C. A quantitative evaluation of protein extracts was carried out with Bradford assay (BIORAD, Hercules, CA, USA). Bovine serum albumin was used as the standard protein for the calibration curve.

### 4.3. Isolation of Protein Complexes by Immunoprecipitation

Lysates from 3xFlag-FTH1 transfected cells and 3xFlag negative control (cells transfected with empty vector) were subjected simultaneously pre-cleared onto Dynabeads^TM^ Protein G (Invitrogen, Waltham, MA, USA); thus, to remove the background of unspecific proteins from cellular extracts. The latter were then subjected to immunoprecipitation protocol by using anti-FLAG M2 (Sigma Aldrich) magnetic beads to isolate the bait, and its putative interactors, as previously reported [39]. Surnatants containing the unbound proteins were removed, and the beads were washed with two different NaCl concentrations (150 and 300 mM) in lysis buffer. Elution was performed using the free 3xFLAG peptide with a concentration of 200 µg/mL for 5 h at 4 °C. Protein samples were precipitated in a mixture of methanol and chloroform and then dried by a Speed-Vac system (Thermo Fisher, Waltham, MA, USA).

### 4.4. Mass Spectrometry Analysis and Protein Identification

Immunoprecipitated protein complexes were fractionated by SDS-PAGE on a 16 × 16 cm, 8–15% gradient acrylamide/bis-acrylamide gel. The latter was stained with Colloidal Blue Coomassie (PIERCE, Hercules, CA, USA), and the excess of the dye was removed by washing with deionized water. Thirty-nine bands were excised from both sample and control lanes and subjected to an in-situ hydrolysis protocol, as previously reported [40]. Each peptide mixture was dried and then suspended again in 10 µL of 0.2% HCOOH (J.T. Baker, Waltham, MA, USA ) and analyzed by LC-MS/MS on an LTQ Orbitrap XL system (Thermo Fisher) equipped with a nano-LC Proxeon nanoEasy-II system. Peptides were fractionated onto a C18 reverse-phase capillary column (5 μm biosphere, 75 μm ID, 200 mm length) working at 250 nL/min flow rate and adopting a linear gradient from 10% to 60% of eluent B (0.2% formic acid, 95% acetonitrile LC-MS Grade) over 69 min. Mass spectrometric analyses were carried out in Data Dependent Acquisition mode (DDA): from each MS scan, spanning from 300 to 1800 m/z, the five most abundant ions were selected and fragmented.

Output data were processed generating. mgf files employed for protein identification procedure in NCBI database according to Mascot licensed software (Matrix Science, Boston, MA, USA). Protein identification was carried out by using 10 ppm as peptides mass tolerance for MS and 0.6 Da for MS/MS search; *Homo sapiens* as taxonomy, carbamidomethyl (C) as fixed modification and Gln→pyro-Glu (N-term Q), Oxidation (M), Pyro-carbamidomethyl (N-term C) as variable modifications. The proteins were identified with at least two significant peptides, overcoming the Mascot score threshold.

### 4.5. Co-Immunoprecipitation

HEK-293T cells were lysated in a Ripa buffer assay (Radioimmunoprecipitation assay) [41,42,43] and incubated overnight with 1 μg of anti-FTH1 antibody or with non-immune serum at 4 °C and then incubated for 1h at 4 °C with Protein A/G plus-agarose. The beads were collected by centrifugation, washed four times with lysis buffer and then loaded onto an SDS 12 % (*w*/*v*) polyacrylamide gel [44].

### 4.6. Western Blotting Analysis

A total of 60 μg of protein extract was resolved on 15% SDS-PAGE and then transferred to a nitrocellulose membrane by electroblotting [32]. Non-specific reactivity was blocked in nonfat dry milk in TTBS 1X [5% (*w*/*v*) milk in TBS 1X (pH 7.4) and 0.1% Tween [20] for 2 h at room temperature. The membrane was incubated with specific primary antibodies anti-FTH1 (sc-376594 Santa Cruz, Santa Cruz, CA, USA), Anti-Flag M2 (F-1804 Sigma Aldrich) and anti-PRDX6 (ab59543 Abcam, Cambridge, UK) overnight at 4 °C. After incubation, and the membranes were washed three times with TTBS 1X for 10 min and incubated with an appropriate horseradish peroxidase-conjugated secondary antibody at room temperature for 1 h. According to the manufacturer’s instructions. The membranes were washed three times with TTBS 1X, and signals were visualized by ECL-Western blot detection reagents (Santa Cruz Biotechnology, Dallas, TX, USA [45]).

### 4.7. Molecular Modeling

Protein Data Bank (PDB) [46] entries 3AJO [47], 5B6M and 5B6N [48] were selected for building our theoretical models of FTH1 and PRDX6 in reduced (PRDX6_r) and sulphinic acid (PRDX6_s) states, respectively. The original PDB 3AJO structure was modified by removing co-crystallized water molecules and Mg^2+^ ions. To identify the most favorable FTH1 sites for iron recognition, FE + 2 probe molecular interaction field (MIF), as implemented in GRID ver. 22d [49], was computed. To obtain a more accurate position of the iron ions, the NPLA GRID keyword was equal to 3, while other parameters were set to the default value. MIF was filtered by means of the GRID MINIM utility using an energy cutoff equal to 10 kcal/mol above the global minimum energy point (−85.27 kcal/mol). Such an approach allowed us to identify two favorable positions for the iron ion with respect to 3AJO PDB entry. Therefore, two iron ions were included in our FTH1 preliminary model. 

Both PRDX6_r and PRDX6_s PDB entries contained, reported as chains A, B and C, three conformationally different structures of the protein. These models were divided by the chain, and each of them was considered a conformer of PRDX6.

To prepare FTH1 and PRDX6 structures for further simulation, these were optimized using MacroModel ver. 11.9 (Schrödinger Release 2018-1: MacroModel, Schrödinger, LLC, New York, NY, USA, 2021). In detail, hydrogen atoms were added, and 10,000 steps of the Polack Ribiere Conjugate Gradient energy minimization method, coupled to the AMBER* force field, were applied. Aqueous environment effects were mimicked by means of the implicit solvation model GB/SA water.

The aim was to investigate the recognition between FHT1 and PRDX6. The resulting optimized structures were submitted to AutoDock Vina v. 1.2.1 molecular docking [50]. Using the MGL-tool ver. 1.5.6, Kollman charge distribution was computed on all protein models. PRDX6_r and _s optimized conformations were considered as receptor models and FHT1 as ligand. Therefore, for each receptor model, an AutoDock Vina simulation was performed, considering a regular box equal to 4,000,752 Å^3^ entirely surrounding targets. For each FHT1·PRDX6 theoretical complex, a maximum of 20 configurations were allowed (num_modes = 30), and the other docking parameters were set to default values. Docking results highlighted 57 and 56 FHT1 poses with respect to PRDX6_r and PRDX6_s, respectively. The affinity of FHT1 against both PRDX6_r and PRDX6_s was estimated by computing the AutoDock Vina average interaction energy values. 

To consider the induced fit phenomena, all theoretical complexes were submitted to the same energy minimization protocol previously reported for the protein models preparation. The most relevant interacting residues were highlighted by analyzing DRY, N1 and O GRID MIFs using a modified GBPM method against all optimized structures [51,52].

### 4.8. Immunofluorescence Assay

For immunofluorescence analysis, HEK-293T and NCI-H460 cells were treated as previously described by Aversa et al. [27]. Briefly, HEK-293T and NCI-H460 cells were incubated with primary antibodies anti-FTH1 (sc-376594 Santa Cruz) and anti-PRDX6 (ab59543 Abcam) overnight at 4 °C. Appropriate secondary antibodies (anti-mouse IgG Alexa Fluor 488 and anti-rabbit IgG Alexa Fluor 555, Thermo Fisher Scientific) diluted in PBS 1X were applied for 1 h at room temperature. After three washes, Nuclear Dapi (1:1000, Invitrogen, Carlsbad, CA, USA) was added for 20 min. The slides were mounted on microscope slides using a mounting solution ProLong Gold antifade reagent (Thermo Fisher Scientific). Images were collected using a Leica TCS SP2 confocal microscopy system (Leica Microsystems, Wetzlar, Germany) [53].

### 4.9. Proliferation Assays

For the proliferation assay 3 × 10^3^ 3xFlag control vector, 3xFlag-PRDX6, 3xFlag FTH1/PRDX6 and 3xFlag-PRDX6/siFTH1 NCI-H460 cells were plated in a 96-well flat bottom tissue culture plate. At 24, 48 and 72 h of culture, 10 μL of 3-(4,5-dimethylthiazol-2-yl)-2, 5-diphenyl tetrazolium bromide (MTT) (Sigma-Aldrich) solution (2 mg/mL) were added per well. After 4 h of incubation, the culture medium was discarded and replaced with 200 μL of isopropanol. Optical density (OD) was read on a multi-well scanning spectrophotometer (ELISA reader) (BIORAD) at 595 nm. The proliferation assay was performed in triplicate.

### 4.10. Wound Healing Assay 

NCI-H460 cells 3xFlag control vector, 3xFlag-PRDX6, 3xFlag FTH1/PRDX6 and 3xFlag-PRDX6/siFTH1 were seeded in 60 mm dishes at a density of 4 × 10^5^. After 24 h, a yellow pipette tip was used to make a scratch. Scratch closure was monitored, and images were captured at 0, 24, 48 and 72 h using a light microscope (using the Leica DFC420 C and Leica Application Suite X Software 3.7.4.23463) (Leica Microsystems). Wound closure was measured by calculating the density of the pixels in the area where the cut was made and expressed as a percentage of wound closure in the area. The percentage of wound closure was calculated by ImageJ64 software.

### 4.11. Statistical Analyses 

Data were presented as the mean ± standard error (SD) of three independent experiments. Statistical and data analysis was carried out using GraphPad Prism 9 software. Statistical differences between samples were assessed by Student’s *t*-test. 

## Figures and Tables

**Figure 1 ijms-23-12987-f001:**
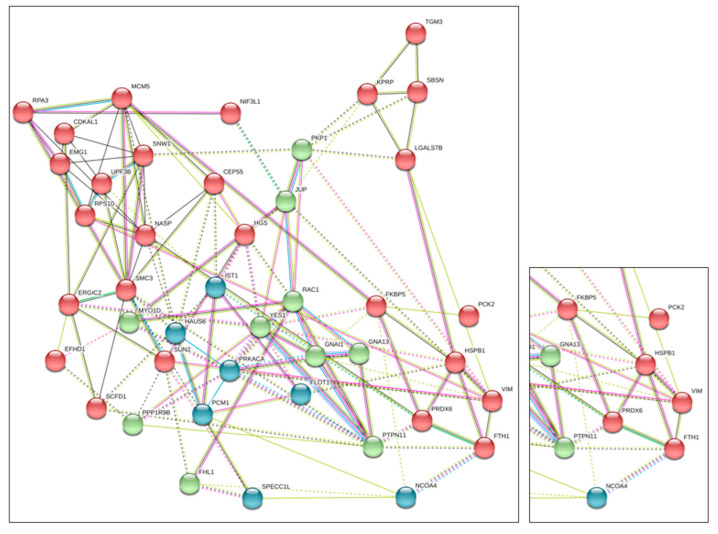
String Interaction Analysis: In silico interaction analysis by String (https://string-db.org/ (accessed on 15 September 2022)) suggests putative interaction of FTH1 and PRDX6.

**Figure 2 ijms-23-12987-f002:**
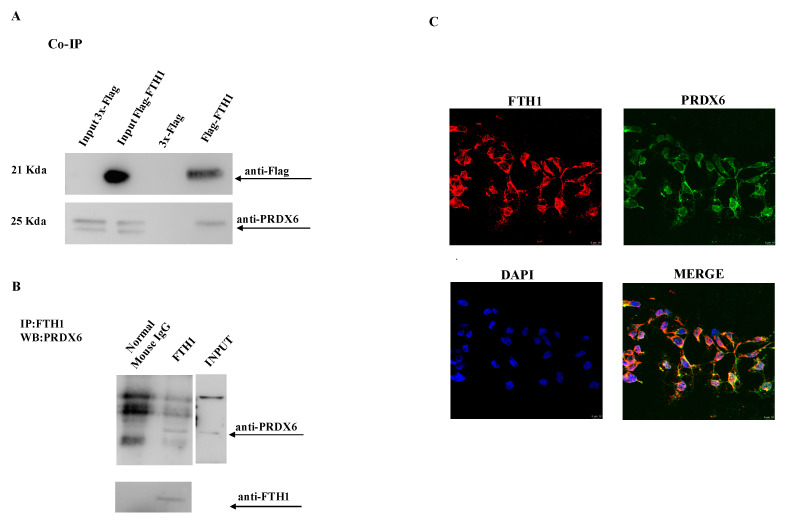
Co-IP and colocalization analysis of the interaction between FTH1 and PRDX6. (**A**) HEK-293T cells were transfected with Flag-FTH1 or a control vector (3x-Flag). Cell lysates were immunoprecipitated with anti-Flag M2 resin and then analyzed by immunoblotting with anti-PRDX6 and anti-Flag antibodies. (**B**) HEK-293T cell lysates were immunoprecipitated with FTH1 antibody or anti-mouse IgG, respectively. Eluates were analyzed by immunoblotting with anti-FTH1 or anti-PRDX6 antibodies. (**C**) HEK-293T cells were grown on a coverslip, fixed with 4% paraformaldehyde and processed for double-label immunofluorescence with anti-FTH1 antibody and anti-PRDX6 antibody. Images were collected using confocal microscopy system (40×). Representative data from one of three experiments.

**Figure 3 ijms-23-12987-f003:**
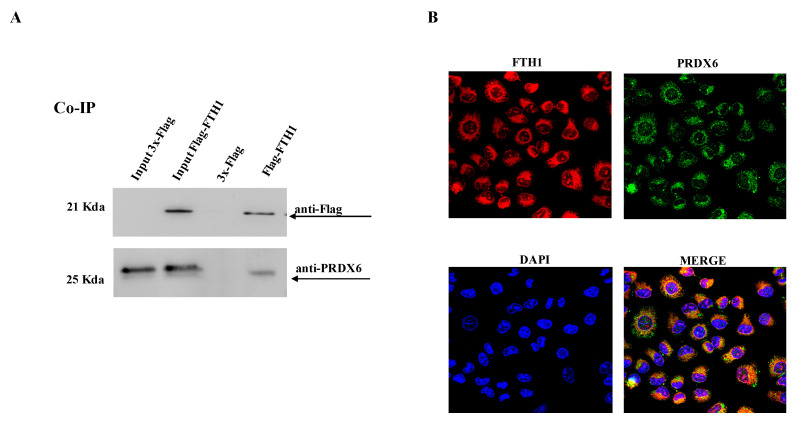
Co-IP and colocalization analysis of the interaction between FTH1 and PRDX6. (**A**) NCI-H460 cells were transfected with Flag-FTH1 or a control vector (3x-Flag). Cell lysates were immunoprecipitated with anti-Flag M2 resin and then analyzed by immunoblotting with anti-PRDX6 and anti-Flag antibodies. (**B**) NCI-H460 cells were grown on a coverslip, fixed with 4% paraformaldehyde and processed for double-label immunofluorescence with anti-FTH1 antibody and anti-PRDX6 antibody. Images were collected using confocal microscopy system (40×). Representative data from one of three experiments.

**Figure 4 ijms-23-12987-f004:**
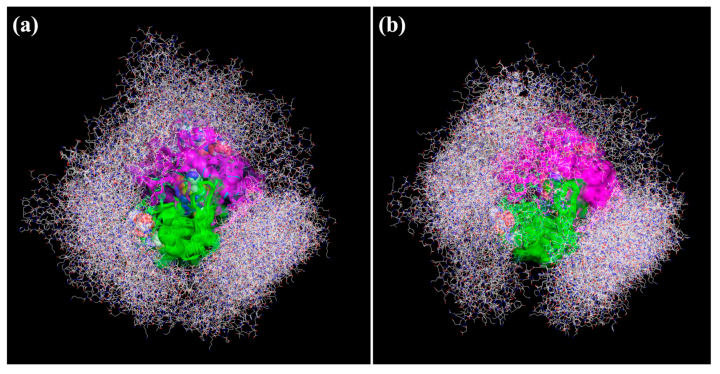
Superimposition of all theoretical complexes between FTH1 and (**a**) PRDX6_r and (**b**) PRDX6_s. Ferritin protein is depicted as wireframe CPK colored, and PRDX6 chains are shown as green and magenta cartoon.

**Figure 5 ijms-23-12987-f005:**
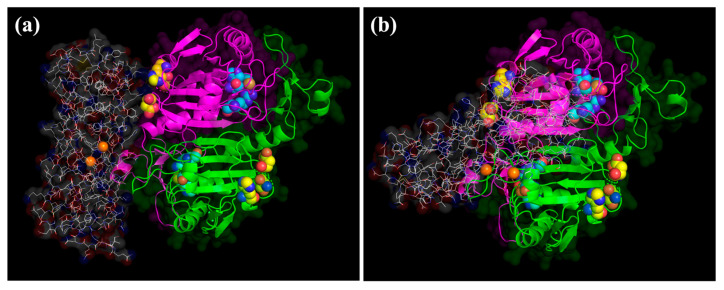
Most stable theoretical complexes of FTH1 with (**a**) PRDX6_r and (**b**) PRDX6_s. Ferritin protein is depicted as wireframe CPK colored, and PRDX6 chains are shown as green and magenta cartoon. PRDX6 residues at peroxidase and phospholipase sites are reported as space-filling yellow carbon or cyan carbon colored, respectively. Ferritin-bound Fe^2+^ ions are shown as orange space-filling.

**Figure 6 ijms-23-12987-f006:**
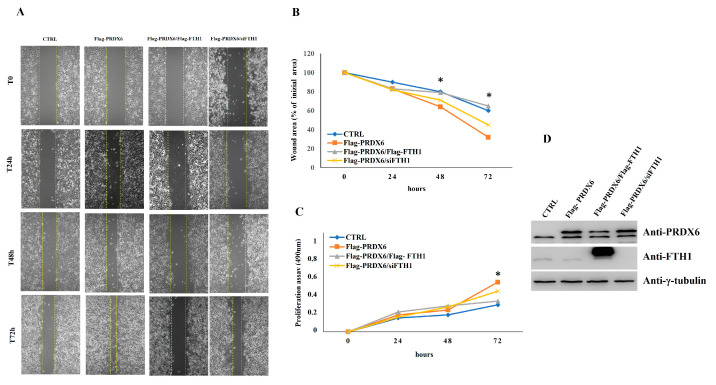
Scratch wound healing and MTT proliferation assays. (**A**) After transfection with control vector, (3xFlag) Flag-PRDX6, Flag-PRDX6/Flag-FTH1 and Flag-PRDX6/siFTH1 confluent NCI-H460 cells monolayers were scratched to induce horizontal migration (magnification of 10×) at 0, 24, 48 and 72 h. (**B**) The wound areas were measured using Image J 1.42q software. The histogram indicates wound area/% of initial area. Value, represent mean ± SD; n = 3; *p*-value Flag-PRDX6_vs._3xFlag, Flag-PRDX6_vs._Flag-PRDX6/Flag-FTH1 and Flag-PRDX6_vs_Flag-PRDX6/siFTH1 is considered not statistically significant at 24 h. Flag-PRDX6_vs_3xFlag, Flag-PRDX6_vs_Flag-PRDX6/Flag-FTH1 is considered statistically significant at 48 and 72 h, * *p* < 0.005; Flag-PRDX6_vs._Flag-PRDX6/siFTH1 is considered not statistically significant at 48 and 72 h. (**C**) NCI-H460 cells, transfected with control vector, Flag-PRDX6, Flag-PRDX6/Flag-FTH1, and Flag-PRDX6/siFTH1 seeded in 24 wells and were cultured for 0, 24, 48 and 72 h, and their proliferation was determined by absorbance at 490 nm. Final results represent mean ± SD of three independent experiments, each performed in triplicate. *p*-value Flag-PRDX6_vs._3xFlag, Flag-PRDX6_vs._Flag-PRDX6/Flag-FTH1 and Flag-PRDX6_vs._Flag-PRDX6/siFTH1 are considered not statistically significant at 24 and 48 h. Flag-PRDX6_vs._3xFlag *p* < 0.05 and Flag-PRDX6_vs_Flag-PRDX6/Flag-FTH1 * *p* < 0.005 at 72 h; Flag-PRDX6_vs._Flag-PRDX6/siFTH1 is considered not statistically significant at 72 h. (**D**) Western Blot analysis was performed to confirm PRDX6 and PRDX6/FTH1 overexpression and siFTH1 silencing. γ-tubulin was used as a loading control.

**Table 1 ijms-23-12987-t001:** FTH1 (left) and PRDX6 (right) most relevant, quartile 1, interacting residues.

FTH1 Residues Interacting to		PRDX6_r	PRDX6_s
*PRDX6_r*	*PRDX6_s*	*Residues Interacting with FTH1*	
Thr5		Gly4	
		Asp9	
Gln14			Asn13
Arg22			Glu15
Tyr32 ^a^		Arg24	
Tyr39		Asp31	
Asp42		Arg53	
	Lys53	Lys56	
Arg63			Tyr89
	Glu64		Asn90
Glu67		Glu109	
Lys71 ^a^		Lys122	
	Asn74^a^		Lys125
Arg79		Lys141	
Gln83		Lys142	
Asp84		Asp158	
Lys86 ^a^		Glu172	
Lys87 ^a,b^		Arg174	
Asp89		Asp180	
Asp91 ^a^		Lys182	
	Lys108	Asp183	
Glu116		Gly184	
Asn125			Asp185
Asn139		Ser186	
Lys143		Lys200	
	Lys157		Lys204
		Lys209	
		Tyr220	

^a^ FTH1 residues interacting to PRDX6 phospholipase site ^b^ FTH1 residues interacting to PRDX6 peroxidase site.

## Data Availability

Data is contained within the article or Appendix A.

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
