# Peer review of "Ferritin Heavy Chain Binds Peroxiredoxin 6 and Inhibits Cell Proliferation and Migration"

_ijms, 2022, doi:10.3390/ijms232112987_

Round 1

Reviewer 1 Report

No comments.

Author Response

See uploaded word file

Reviewer 2 Report

The paper of Di Sanzo et al is devoted to study the functional role of protein-protein interaction of Ferritin Heavy Chain (FHC1) with Peroxiredoxin 6 (Prdx1). The authors have used proteomic approach to identify the spectrum of proteins interacting with FHC1; Prdx1 was identified as one of the interactors. The authors have carried out a number of experiments to verify the interaction and demonstrating that FTH1 inhibits Prdx1-mediated proliferation and migration.

To my opinion, the number of points should be addressed:

1)    Figure 2a and B, and 3a: the equal quantity of inputs of Prdx1 should be shown at the each of Co-IP figures;

2)    To improve the functional significance of FHC1-Prdx1 interaction, the experiment with knock-down of FHC1 should be carried out;

3)    The paper would benefit from one more experiment demonstrating the FHC1-mediated down-regulation of Prdx1-induced proliferation. For instance, colony-formation assay which is rather simple and robust;

4)    What about other cells? The authors may carry out experiment showing in Figure 6 with the other cell lines of lung adenocarcinoma or other types of cancer to demonstrate that possibly the functional significance of this interaction does not depend on the cell line

Author Response

See uploaded word file

Round 2

Reviewer 2 Report

Authored have addresed main reviewers issues